# Effect of Phlorofucofuroeckol A and Dieckol Extracted from *Ecklonia cava* on Noise-induced Hearing Loss in a Mouse Model

**DOI:** 10.3390/md19080443

**Published:** 2021-08-01

**Authors:** Hyunjun Woo, Min-Kyung Kim, Sohyeon Park, Seung-Hee Han, Hyeon-Cheol Shin, Byeong-gon Kim, Seung-Ha Oh, Myung-Whan Suh, Jun-Ho Lee, Moo-Kyun Park

**Affiliations:** 1Department of Otorhinolaryngology-Head and Neck Surgery, Seoul National University College of Medicine, Seoul National University Hospital, Seoul 03080, Korea; junwoo1207@naver.com (H.W.); jkmk928@gmail.com (M.-K.K.); vivihelia@gmail.com (S.P.); hee92h@nate.com (S.-H.H.); byeonggone@naver.com (B.-g.K.); shaoh@snu.ac.kr (S.-H.O.); drmung@naver.com (M.-W.S.); junlee@snu.ac.kr (J.-H.L.); 2Interdisciplinary Program in Neuroscience, College of Natural Sciences, Seoul National University, Seoul 08826, Korea; 3CEWIT Center for Systems Biology, State University of New York, Incheon 21985, Korea; gerudah@gmail.com; 4Sensory Organ Research Institute, Medical Research Center, Seoul National University, Seoul 03080, Korea; 5Wide River Institute of Immunology, College of Medicine, Seoul National University, Hongcheon 25159, Korea

**Keywords:** noise, hearing loss, dieckol, PFF-A, antioxidant

## Abstract

One of the well-known causes of hearing loss is noise. Approximately 31.1% of Americans between the ages of 20 and 69 years (61.1 million people) have high-frequency hearing loss associated with noise exposure. In addition, recurrent noise exposure can accelerate age-related hearing loss. Phlorofucofuroeckol A (PFF-A) and dieckol, polyphenols extracted from the brown alga *Ecklonia cava*, are potent antioxidant agents. In this study, we investigated the effect of PFF-A and dieckol on the consequences of noise exposure in mice. In 1,1-diphenyl-2-picrylhydrazyl assay, dieckol and PFF-A both showed significant radical-scavenging activity. The mice were exposed to 115 dB SPL of noise one single time for 2 h. Auditory brainstem response(ABR) threshold shifts 4 h after 4 kHz noise exposure in mice that received dieckol were significantly lower than those in the saline with noise group. The high-PFF-A group showed a lower threshold shift at click and 16 kHz 1 day after noise exposure than the control group. The high-PFF-A group also showed higher hair cell survival than in the control at 3 days after exposure in the apical turn. These results suggest that noise-induced hair cell damage in cochlear and the ABR threshold shift can be alleviated by dieckol and PFF-A in the mouse. Derivatives of these compounds may be applied to individuals who are inevitably exposed to noise, contributing to the prevention of noise-induced hearing loss with a low probability of adverse effects.

## 1. Introduction

Hearing loss impairs individuals’ communication, comprehension, and quality of life. One of the well-known causes of hearing loss is noise [1]. Noise-induced hearing loss (NIHL) is one of the most common occupational diseases worldwide [2]. Based on the national health and nutrition examination surveys, approximately 12.8% of Americans between the ages of 20 and 69 years have noise-induced hearing threshold shift. [3]. Furthermore, NIHL in teenagers has gathered increasing attention [4]. Considering the substantial medical costs, NIHL is an important social, clinical, and economical issue [5]. Two types of NIHL are known: permanent threshold shift (PTS) and temporary threshold shift (TTS). Because of the reversibility of hearing after TTS, previous studies have considered it less important than PTS. However, a recent study has suggested that TTS can induce synaptopathy, thus accelerating age-related hearing loss [6]. As a result of these considerations, the prevention of TTS has been gaining increased attention [7].

Conventionally, mechanical trauma was thought to be the main cause of NIHL. The total amount of noise is determined by the sound pressure of noise and the duration of exposure over time. Therefore, a low level of noise over a long period of time may have the same damage as a higher level of noise over a short period of time. Recommended precautions for NIHL include avoiding or minimizing exposure to prolonged or loud noise [8]. However, these preventive measures are not applicable to populations that cannot avoid or reduce noise exposure, such as construction workers and soldiers. Recent studies revealed that NIHL is caused by reactive oxygen species (ROS) evoked by excessive noise stimulation. Various studies on the use of antioxidants to prevent NIHL are in progress [8,9]. Although preventive treatments must be applied before the development of NIHL, prescribing preventive medication without any certainty about whether NIHL will develop is risky considering the probable adverse effects of drugs. However, food components can be used with lower risk.

Brown algae are commonly used as dietary supplements and herbal remedies in Asian countries. Among the brown algal species, *Ecklonia cava* produces unique polyphenols named eckols. Although *E. cava* produces various potentially medicinal compounds, such as common sterols (fucosterol, cholesterol, and ergosterols) and phlorotannins (eckol, dieckol, and phlorofucofuroeckol A (PFF-A); Figure 1), eckol and its derivatives are reportedly responsible for the major medicinal properties of this brown alga [10,11]. Recently, various in vitro and in vivo studies indicated that eckols possess a wide spectrum of bioactivities including matrix metalloproteinase inhibitory, protease inhibitory, cytoprotective, anti-inflammatory, and antioxidant effects [12,13,14]. A previous study has shown a considerable protective effect of a purified polyphenolic extract from *E. cava* against TTS. This extract consists of eckols including dieckol and PFF-A [15]. In this study, we investigated the protective effect of dieckol and PFF-A against TTS in a mouse model of NIHL.

## 2. Results

### 2.1. Radical Scavenging Activity of Dieckol and PFF-A

The radical-scavenging activity of PFF-A and dieckol increased in a dose-dependent manner (Figure 2). These results suggest that dieckol and PFF-A lowered ROS levels. 

### 2.2. ABR Threshold Shifts after Noise Exposure

Saline control showed that ABR thresholds at click and 4 and 16 KHz increased sharply after noise exposure and then gradually decreased at 1 and 3 days. However, the hearing did not return to normal 1 or 3 days after noise exposure. The threshold shift was larger in 16 kHz than click and 4 kHz. 

PFF-A (10 mg/kg) showed a lower threshold shift than saline at click 1 day after noise exposure. However, it was not statistically significant. The ABR threshold shifts 1 day after noise exposure were significantly smaller in the high-dose PFF-A (100 mg/kg) with noise group than in the saline with noise group at click stimuli (*p* = 0.035, t = 2.329 df = 14). The ABR threshold shifts 1 day after noise exposure were significantly smaller in the high-dose PFF-A (100 mg/kg) with noise group than in the saline with noise group at 16 kHz stimuli (*p* = 0.018, t = 2.687 df = 14). Dieckol (10 mg/kg) and high-dieckol (100 mg/kg) groups had lower threshold shifts than the Saline + Noise group at click and 16 kHz, but it was not statistically significant. The ABR threshold shifts in the dieckol with the noise group were significantly smaller than those in the saline with noise group immediately after noise exposure at 4 kHz (*p* = 0.042, t = 2.250 df = 13) (Figure 3). ABR amplitudes and latencies after dieckol or PFF-A treatment were not significantly different from those after saline treatment in all waves at 90 dB stimulation (data not shown).

### 2.3. Hair Cell Phalloidin Staining and Counts

Survival at 1 and 3 days after noise exposure was measured. Significant outer hair cell (HC) survival was observed only in the apical turn sections on day 1 of the High-PFF-A group (*p* = 0.019, t = −2.793 df = 10). In the basal turn, the High-PFFA group showed better survival than the PFF-A group at 3 days after noise exposure (*p* = 0.032 t = −2.276 df = 24) (Figure 4). All three rows of outer HCs from control mice showed no missing hair cells.

## 3. Discussion

This study suggests that PFF-A and dieckol may alleviate the noise-induced temporary threshold shift and HC damage in a mouse model. 

The hearing threshold shift was greatest between 1 and 3 days, but OHC loss was observed on days 1 and 3. Therefore, the animal model in this experiment is likely to be a combined threshold shift model rather than a pure TTS model. Broadband (0.2–70 kHz) white noise, which had its peak at 10 kHz, was used to induced hearing loss. Therefore, hearing threshold shifts at 4 and 16 kHz were measured in this study. Click stimulation for hearing threshold shift was used because click stimulation includes the wide frequency on the cochlea. Therefore, the waveform by click stimulation shows the bigger and clearer than other tone burst stimulation. High-frequency stimulation, including 32 kHz, could be more correlated with hair cell loss in basal turn. In a previous study, we used purified polyphenolic extract from *Ecklonia cava* (100 mg/kg) and observed protective effect on noise-induced hearing loss [15]. Purified polyphenolic extract from *Ecklonia cava* includes the 16.85% of dieckol and 3.5% of PFF-A. Therefore, the same (100 mg/kg) and lower dosage (10 mg/kg) of dieckol and PFF-A for this experiment were used. 

PFF-A could be a more potent candidate for preventive medication than dieckol because PPF-A shows a stronger radical scavenging activity than dieckol at the same concentration. Dieckol showed a lower threshold shift and hair cell survivable than PFF-A. In addition, high dieckol (100 mg/kg) did not show a greater threshold shift than dieckol (10 mg/kg) at 4 kHz after noise. Paudel et al. reported that PPF-A (phloroglucinol pentamer) could be a more potent antioxidant than dieckol (phloroglucinol hexamer) [16]. The functional effect of PFF-A at tested G-protein coupled receptors is higher than that of dieckol. Dieckol has a higher number of hydroxyl groups, but the structure or orientation of PFF-A could reach the core active site cavity of receptors. 

Noise generates ROS in the inner ear, which produce several compounds by peroxidation of polyunsaturated fatty acids [17]. 8-Isoprostaglandin F2a (8-iso-PGF2a), one of these compounds, is a powerful vasoconstrictor that reduces cochlear blood flow and induces cochlear ischemia [18,19]. Subsequently, cochlear ischemia causes excessive release and accumulation of glutamate from the inner hair cells, leading to glutamate excitotoxicity [20,21]. Cochlear ischemia also reduces energy supply to the stria vascularis, leading to decreased endocochlear potential and to HC swelling [22,23]. These processes ultimately lead to dysfunction of HCs and cochlear afferent neurons, resulting in TTS. 

Antioxidants protect against NIHL [24,25]. Diverse antioxidants, including N-acetyl-L-cysteine (NAC), acetyl-L-carnitine (ALCAR), 4-hydroxy alpha-phenyl-tert-butylnitrone (4-OHPBN), salicylate, 2,4-disulfonyl a-phenyl tertiary butyl nitrone (HPN-07), HK-2, and others, show protective effect against noise-induced damage in the cochlea [26,27,28,29,30]. 

PFF-A, a bioactive component of polyphenols extracted from *E. cava*, has been reported to have antioxidant, anti-inflammatory, anti-allergic, and anti-cancer effects [31,32,33,34,35]. PFF-A has an antioxidant effect via scavenging of ROS and eliminating or reducing the ROS production as such. PFF-A also shows an anti-inflammatory effect via inhibition of nitric oxide and prostaglandin E2 (PGE2) synthesis via downregulation of the levels of iNOS and COX-2 proteins [34]. 

Dieckol, one of the major bioactive components among polyphenols extracted from *E. cava*, has well-documented antioxidant, cytoprotective, and anti-inflammatory effects [36,37,38]. Dieckol is considered to inhibit ROS activity by removing ROS [31,36,39] and suppressing ROS formation through upregulation of antioxidant enzymes including superoxide dismutase (SOD) and glutathione peroxidase [37], and downregulation of pro-inflammatory enzymes, such as nitric oxide synthase and cyclooxygenase-2 (COX-2) [39].

Considering the pathophysiology of TTS, an effective preventative treatment must be applied before noise exposure. However, potential side effects of drugs make it difficult to administer medication without any certainty on whether the disease will develop. The advantage of using PFF-A and dieckol over the use of medications is that these natural extracts from *E. cava* are as safe as food ingredients. Furthermore, a representative *E. cava* polyphenol extract that contains PFF-A and dieckol was approved by the United States Food and Drug Administration as a new dietary ingredient in 2008 (FDA-1995-S-0039-0176). Therefore, PFF-A and dieckol may be administered for the prevention of TTS to those who are inevitably exposed to noise, with less possibility of adverse effects than in the case of medications. 

In this study, mice were administered dieckol and PFF-A via intraperitoneal injection. However, for humans, dieckol and PFF-A would be administered via oral intake. According to our unpublished bioinformatic analysis that was performed during the study, dieckol and PFF-A have low gastrointestinal absorbability and poor water solubility. Although they are potent antioxidant agents and safe materials, additional measures are required before the attempts of oral administration to achieve effective gastrointestinal absorption and significant bioavailability. A report on nanoencapsulation of various polyphenols, including ellagitannins, curcumin, oleuropein, and hydroxytyrosol, has revealed increased gastrointestinal absorption of these polyphenols [40]. In the bioinformatic analysis using SwissADME, both dieckol and PFF-A have high values of WLOGP, which represent lipid solubility, suggesting their high lipid solubility and possible blood–brain barrier (BBB) permeability. However, both compounds also have high values of topological polar surface area. Thus, both dieckol and PFF-A likely show minimal BBB permeability, implying that the effects of both compounds on the inner ear after they enter the systemic circulation may be negligible. To overcome this second obstacle, extra measures to increase BBB permeability are necessary. A study with fluorescein isothiocyanate (FITC)-labeled dieckol and a rhodamine B-labeled dieckol reported effective BBB penetration of these dieckol forms in rats [41]. Further studies are required to establish an effective delivery method of orally applied dieckol and PFF-A into the inner ear. 

In vivo study of radical-scavenging activity of PFF-A and dieckol would provide evidence for their antioxidant effects. Further in vivo studies performed in humans would be helpful to demonstrate the protective effect of PFF-A and dieckol against TTS or PTS. 

## 4. Materials and Methods

### 4.1. Preparation of PFF-A and Dieckol

Purified dieckol and PFF-A were prepared and kindly supplied by BotaMedi, Inc. (Jeju, Korea). Briefly, the whole plant of *E. cava* was collected off the coast of Jeju Island, Korea. Dried *E. cava* powder was extracted with 70% aqueous ethanol and then partitioned between water and ethyl acetate. The ethyl acetate fraction was subjected to octadecylsilyl (ODS) column chromatography followed by gel filtration in a Sephadex LH-20 column equilibrated with methanol. Final purification of individual compounds was accomplished by HPLC (Waters Spherisorb S10 ODS2 column (20 × 250 mm); eluent, 30% methanol; flow rate, 3.5 mL/min) to isolate dieckol (98.5 wt%) and PFF-A (98.0 wt%).

### 4.2. 1,1-Diphenyl-2-picrylhydrazyl Assay

Dieckol and PFF-A were diluted in distilled water to obtain the experimental concentrations (0, 1, 12.5, 25, 50, 100, and 200 μM). 1,1-Diphenyl-2-picrylhydrazyl (DPPH) powder (Sigma) was dissolved in 95% methanol to create a 1 M stock. PFF-A, dieckol, and 50% methanol were aliquoted into individual wells of a 96-well plate. The DPPH solution was then added to each well. PFF-A and dieckol were allowed to react with DPPH solution in the dark for 30 min at room temperature. Absorbance was measured at 540 nm using a microplate reader. Two wells were used for each concentration, and the experiment was repeated twice.

### 4.3. Noise Exposure

Six-week-old male C57BL/6 mice weighing 18–20 g were purchased from Koatech Inc. (Pyeongtaek, Korea). The mice were fed a standard commercial diet and housed in a facility with an ambient temperature of 20–22 °C and a relative humidity of 50 ± 5% under 12/12 h day/light cycle conditions. All of the animal experiments described here were approved by the Institutional Animal Care and Use Committee of Seoul National University Hospital (Seoul, Korea; 18-0025-C1A0), which is endorsed by the Association for the Assessment and Accreditation of Laboratory Animal Care International. 

A total of 46 mice were randomly assigned to the following groups: Saline (n = 6), Saline + Noise (n = 8), PFF-A (10 mg/kg) + Noise (n = 8), High PFF-A (100 mg/kg) + Noise (n = 8), Dieckol (10 mg/kg) + Noise (n = 8), and High- dieckol (100 mg/kg) + Noise (n = 8) (Figure 5). Mice were administered dieckol and PFF-A via intraperitoneal injection. The mice were treated on 3 consecutive days starting 1 day before the noise exposure. The mice were exposed to 115 dB SPL of noise one single time for 2 h. 

Mice were anesthetized via an intramuscular injection before noise exposure using a mixture of 40 mg/kg Zoletil (Zoletil 50; Virbac, Bogotá, Colombia) and 10 mg/kg xylazine (Rumpun; Bayer-Korea, Seoul, Korea). Each mouse was placed in a separate wire cage to avoid uneven noise exposure. Experiment was performed in a specially designed acrylic box in a sound-attenuating laboratory booth with an electromagnetic shield. The mice were exposed to broadband white noise at 115 dB SPL using a 2446-J compression driver (JBL Professional, Los Angeles, CA, USA) with an MA-620 power amplifier (Inkel, Incheon, Korea). 

### 4.4. Auditory Brainstem Response Recordings

Audiometry was conducted 4 h after the termination of noise exposure to perform as stable measurements as possible (Figure 2). Then, mice were divided into 2 equal subgroups (n = 4 each). Hearing was evaluated in one subgroup at 1 day after the noise exposure (Day 1 group) and in the other subgroup at 3 days after the noise exposure (Day 3 group). 

The mice were anesthetized and placed in sound-attenuating booths. The sound stimuli applied were tone-bursts of clicks (4 and 16 kHz, duration, 1562 μm; CoS shaping, 21 Hz). High-frequency software (ver. 3.30; Intelligent Hearing Systems, Miami, FL, USA) and high-frequency transducers (HFT9911-20-0035; Intelligent Hearing Systems) were used to measure the auditory brainstem response (ABR). Before obtaining the electroencephalography signal, the impedance between the electrodes was evaluated to determine whether this was less than 2 kΩ. Responses to the signal were amplified approximately 100,000-fold and band-pass filtered (100–1500 Hz). The intensity of the stimuli covered from 20 to 90 dB SPL in 5 dB increments; 512 sweeps in total were averaged at each intensity level. Additional ABRs were measured at 4 h, day 1, and day 3 after noise exposure. The ABR threshold was defined as the lowest stimulus intensity level that produced an evident waveform for wave II or IV. Latency of waves II–IV and amplitude of waves II and IV at 90 dB stimuli were also analyzed and compared between experimental groups. 

### 4.5. Outer Hair Cell Staining

All mice were sacrificed under anesthesia on the day of their hearing evaluation. The cochlea was detached from the temporal bone and fixed in 4% paraformaldehyde solution for 24 h at 4 °C. The thin layer of laminar bone surrounding the cochlea was carefully decorticated using a 2 mm diameter diamond burr drill (Strong 90; Saeshin Precision, Daegu, Korea) and the thinned laminar bone was then removed using a conventional 1 mm syringe needle. Next, the stria vascularis and Reissner’s membrane were removed. Phalloidin was used to stain F-actin, and the photostable orange fluorescent Alexa Fluor 546 dye was used to visualize the cuticular plate and stereocilia within the hair cells (HCs). After surface preparation, the isolated spiral structure of the organ of Corti was incubated in a solution containing 0.3% Triton X-100 and Alexa Fluor 546 phalloidin (1:100 dilution; Invitrogen, Carlsbad, CA, USA) for 60 min at room temperature in a light-proof box [29]. The sample was separated into three segments using Vannas capsulotomy scissors (E-3386; Karl Storz SE & Co. KG, Tuttlingen, Germany). The first complete turn from the top of the organ of Corti was named apical section, the second complete turn was named middle section, and the final half-turn was named basal section. Then, the sections were mounted on a slide using ProLong Gold Antifade mountant (P36930; Invitrogen). Hair cells were measured in five different areas of each section with the images obtained using a STED CW confocal laser scanning system (Leica, Wetzlar, Germany).

### 4.6. Statistical Analyses

All data were expressed as the mean ± standard error of the mean and analyzed using SPSS software (ver. 25; IBM, Armonk, NY, USA). An F-test was performed to determine whether the levels of variation within the groups were equal. After the F-test, the data were analyzed using Student’s t-tests to identify significant differences between groups. A *p*-value of < 0.05 was considered statistically significant.

## 5. Conclusions

Our investigation of the effect of PFF-A and dieckol against TTS in mice revealed that these compounds isolated from *E. cava* prevented TTS through their antioxidant activity. Since these compounds are dietary ingredients approved by USFDA, they may be used with less possibility of adverse effects than in the case of medications. 

## Figures and Tables

**Figure 1 marinedrugs-19-00443-f001:**
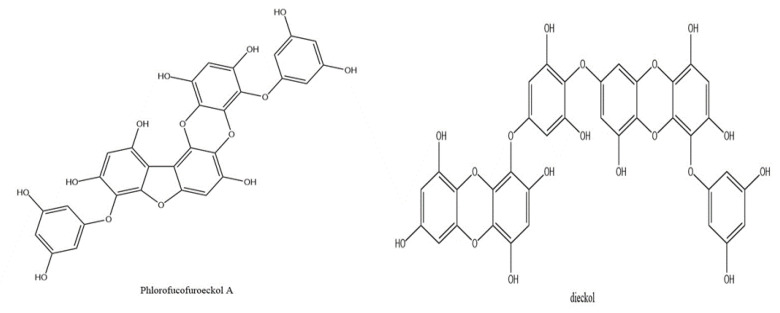
The chemical structures of phlorofucofuroeckol A and dieckol.

**Figure 2 marinedrugs-19-00443-f002:**
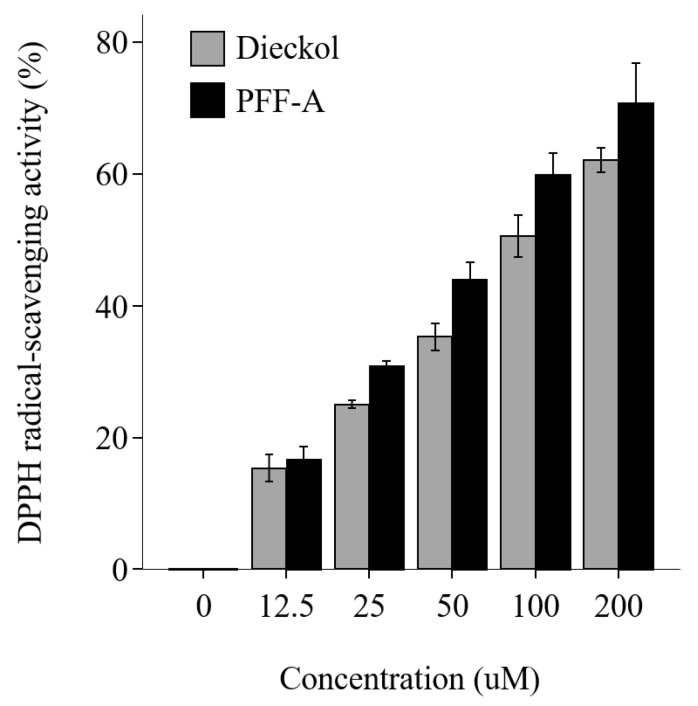
Antioxidant effect of dieckol and phlorofucofuroeckol A (PFF-A), measured as 1,1-diphenyl-2-picrylhydrazyl (DPPH) radical-scavenging activity. Bars represent the mean ± SD.

**Figure 3 marinedrugs-19-00443-f003:**
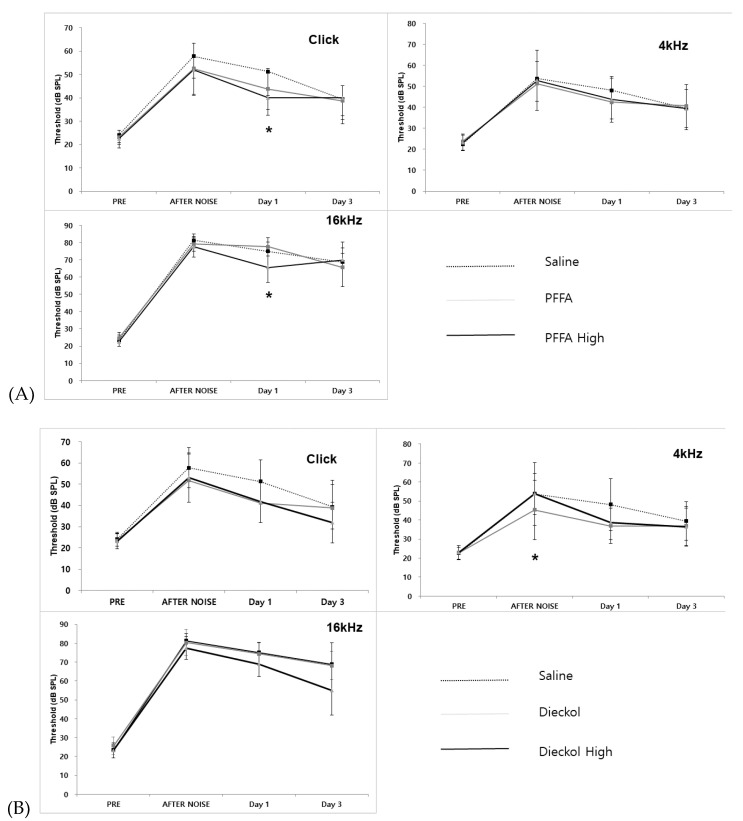
Hearing changes after noise exposure in phlorofucofuroeckol A (PFF-A) and dieckol treatment. Hearing was measured at 4 h and 1 and 3 days after noise exposure at click and 4 and 16 kHz. (**A**) PFFA and High PFF-A treatment; the high-PFF-A group had a significantly lower threshold shift than the Saline + Noise group at click stimuli (*p* = 0.015) and 16 kHz (*p* = 0.018) 1 day after noise exposure. (**B**) Dieckol and High-Dieckol treatment. The Dieckol group showed less threshold shift than the Saline + Noise group at immediately after noise exposure at 4 kHz (*p* = 0.042) (* *p* < 0.05).

**Figure 4 marinedrugs-19-00443-f004:**
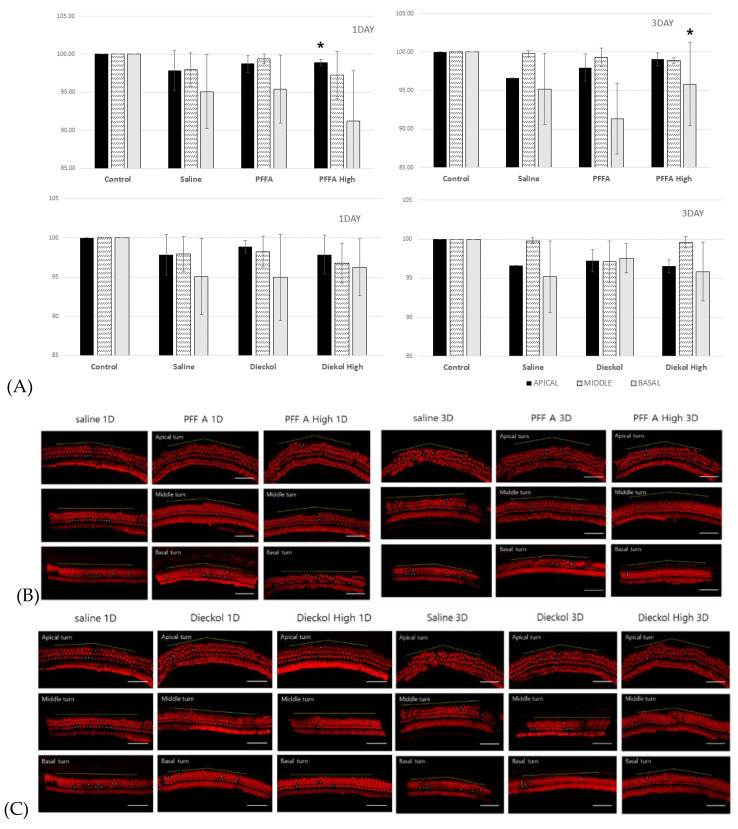
Hair cell (HC) survival and phalloidin staining of outer hair cells. (**A**) Survival rates of outer HCs in each turn section. The surviving HCs per 200 μm along the length of the cochlea in the basal, middle, and apical turn sections were counted. (* *p* < 0.05) (**B**) Fluorescence staining of outer HCs after phlorofucofuroeckol A (PFF A) treatment. (**C**) Fluorescence staining of outer HCs after dieckol treatment. Scale bars are 50 μm. Asterisks indicate the positions of lost HCs. The blue line along the HC line indicates the length of 200 μm.

**Figure 5 marinedrugs-19-00443-f005:**
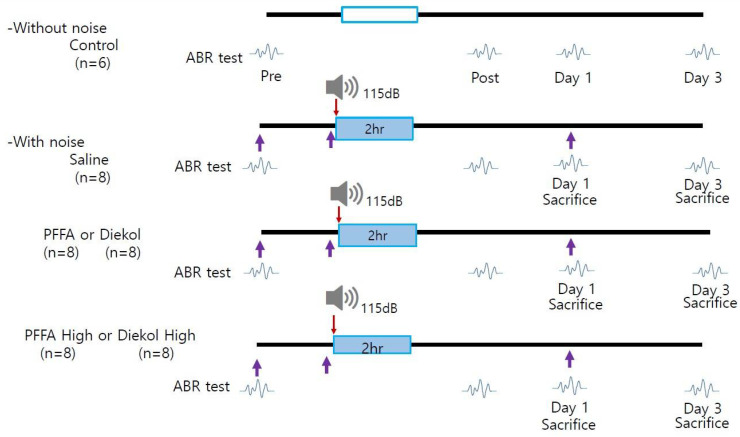
Noise exposure protocol. Empty box indicates control condition without noise exposure. ABR: auditory brainstem response; PPF-A: phlorofucofuroeckol A. Purple arrow: intraperitoneal injection.

## Data Availability

The datasets generated during and/or analyzed during the current study are available from the corresponding author on reasonable request.

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
