# Peer review of "Effect of Phlorofucofuroeckol A and Dieckol Extracted from *Ecklonia cava* on Noise-induced Hearing Loss in a Mouse Model"

_marinedrugs, 2021, doi:10.3390/md19080443_

Round 1
Reviewer 1 Report
marinedrugs-1288636-peer-review-v1
The manuscript entitled "Effect of Phlorofucofuroeckol A and Dieckol Extracted from Ecklonia Cava on Noise-induced Hearing Loss in a Mouse Model" addresses a relevant and under-published topic, so I am of the opinion that it should be accepted for publication after the introduction of corrections. then suggested
Corrections needed:
line 17/18 - Ecklonia cava (in italics)
line 55 - Ecklonia cava (in italics)
line 56 - E. cava (in italics)
line 60 - in vitro ... in vivo (in italics)
line 63 - E. cava (in italics)
line 130 - E. cava (in italics)
line 137 - E. cava (in italics)
line 146 - E. cava (in italics)
line 177 - Briefly, the whole plant of Ecklonia cava (Ochrophyta, Phaeophyceae) was collected off the coast of Jeju Island,
line 194 - 18–20 g were ...
line 196 - humidity of 50 ± 5% under
line 203 - Dieckol (10 mg/kg)
line 263 - E. cava (in italics)
line 274 - Ecklonia cava (in italics)
line 359 - Ecklonia stolonifera (in italics)
line 364 - Ecklonia stolonifera (in italics)
line 366 - In Vitro (in italics)
line 375 - Ecklonia cava (in italics)
Author Response
line 17/18 - Ecklonia cava (in italics)
-> Due to changes in the manuscript, currently the word is in line 16/17. I rewrote the word in italics.
line 55 - Ecklonia cava (in italics)
-> Due to changes in the manuscript, currently the word is in line 54. I rewrote the word in italics.
line 56 - E. cava (in italics)
-> Due to changes in the manuscript, currently the word is in line 55. I rewrote the word in italics.
line 60 - in vitro ... in vivo (in italics)
-> Due to changes in the manuscript, currently the word is in line 59. I rewrote the words in italics.
line 63 - E. cava (in italics)
-> Due to changes in the manuscript, currently the word is in line 62. I rewrote the word in italics.
line 130 - E. cava (in italics)
-> I rewrote the word in italics
line 137 - E. cava (in italics)
-> I rewrote the word in italics
line 146 - E. cava (in italics)
-> I rewrote the word in italics
line 177 - Briefly, the whole plant of Ecklonia cava (Ochrophyta, Phaeophyceae) was collected off the coast of Jeju Island,
-> I rewrote the word in italics
line 194 - 18–20 g were ...
-> I inserted a space between "18-20" and "g". 0
line 196 - humidity of 50 ± 5% under
-> I erased the "%" behind the number 50
line 203 - Dieckol (10 mg/kg)
-> I inserted a space between number 10 and "mg/kg".
line 263 - E. cava (in italics)
-> I rewrote the word in italics
line 274 - Ecklonia cava (in italics)
-> I rewrote the word in italics
line 359 - Ecklonia stolonifera (in italics)
-> I rewrote the word in italics
line 364 - Ecklonia stolonifera (in italics)
-> I rewrote the word in italics
line 366 - In Vitro (in italics)
-> I rewrote the word in italics
line 375 - Ecklonia cava (in italics)
-> I rewrote the word in italics
Reviewer 2 Report
Woo et al. reports the potential otoprotective effect of two polyphenols extracted from a brown algae. While the study is interesting, there are multiple concerns that question the conclusions of the study. They are listed below.
1) Studies from other labs (Ohlemiller, Hirose, Kaur, Liberman etc.) have demonstrated that 112dB SPL or higher intensity of noise for 2 hrs. duration will produce a TTS in the C57BL/6 strain. In this study, the authors have used 115dB SPL noise for 2 hours to study TTS. At this level, in this strain, days 1-3 are likely to be CTS rather than TTS.
2) The noise exposure was broadband white noise. The bandwidth used has not been mentioned. Given that the animals were exposed to broadband noise, why was pure tone ABR performed only at 2 frequencies?
3) The authors report statistical significant difference in the ABR thresholds for the high PFF-A group on day 1. From the graphs in figure 3, it looks like the error bars are overlapping. This raises question about the statistical test result being reported. The authors also seem to selectively show the positive or negative error bars in the figures.
4) For the hair cell counts, it's not clear what PFF groups are being compared to. Is it the control group or saline group? Again the bar graphs in Fig. 4 seem to overlap making any statistical significance being reported suspect.
Author Response
1) We agree with your opinion. So, our model is not a pure TTS model. We added your comments to the discussion as following.
We observed OHC loss on day 1 and day3. So, the animal model in this experiment is likely to be a combined threshold shift model rather than a pure TTS model.
2) Broadband (0.2-70kHz) white noise, which had a peak at 10 kHz, was used to induce hearing loss.
We measured hearing threshold shift at the click, 4, and 16 kHz in this study because we observed more reliable waves with these stimulation frequencies. We used click because Click evoked ABR is highly correlated with hearing sensitivity in the wide frequency range. In addition, the waveform by click stimulation shows the bigger and clearer than ton burst stimulation. So, it is suitable to measure the latencies and to use references.
3) As you mentioned, some error bars are overlapping. So, we selectively show the positive or negative error bars to clearly see the size of each data error. We changed the figures as you recommended, which shows both error bars. The statistical test is significant because the error bar is not much high in control groups.
4) We found the typos and changed the figures. We clearly defined what groups are being compared to.
Significantly less HC loss was observed in the high-dose PFF-A (100 mg/kg) with noise group compared to the control group on day 3 after noise exposure in the apical turn section.
Reviewer 3 Report
The objective of this manuscript was to verify the prophylactic effect of dieckol and PFF-A against temporary threshold shift in mice. The findings suggest that dieckol and PFF-A, when used as preventive agents, may play an important role in reducing the incidence of TTS.
About limitations the authors should modify statistical analysis as well as manuscript formatting.
About strenghts the authors explored the topic and they obtained the purpose of the study.
The methods used are sufficiently documented and allow replication studies. Results obtained are well explained and data interpretation is also correct. Conclusions are consistent with the evidence and arguments presented.
I will recommend the acceptance of this manuscript after these minor corrections:
P values should appear in the legends of figure and they should not be in the manuscript. The authors should correct this inaccuracy;
About references the authors should place before the punctuation; for example [1], [1–3] or [1,3] as required by editorial guidelines. They should replace [12] [13,14] with [12-14], [16] [17] with [16,17] and [26] [27] [28,29] [30] with [26-30].
Author Response
Thank you for the review and comments.
1. P values should appear in the legends of figure and they should not be in the manuscript. The authors should correct this inaccuracy;
- I erased the p-values written in the line 85, 86, 89 of the manuscript and inserted the p-values in the legends of figure 3.
- I erased the p-values written in the lines 100, 101 of the manuscript and inserted them in the legends of figure 4.
2. About references the authors should place before the punctuation; for example [1], [1–3] or [1,3] as required by editorial guidelines. They should replace [12] [13,14] with [12-14], [16] [17] with [16,17] and [26] [27] [28,29] [30] with [26-30].
- I rewrote [12][13,14] into [12-14] and [16][17] into [16,17]
- I replaced [26] [27] [28,29] [30] with [26-30].
Thank you very much.
Reviewer 4 Report
In this study of the protective effects of dieckol and PFF-A on TTS in mice, 48 C57BL/6 mice were randomly assigned to 6 groups: Saline control (n = 6); Saline + Noise (n = 8), PFF-A (10mg/kg) + Noise (n = 8), High PFF-A (100 mg/kg) + Noise (n = 8), Dieckol (10mg/kg) + Noise (n = 8), and High- dieckol (100 mg/kg) + Noise (n = 8). Noise groups were exposed to white noise at 115 dB re: 20 µPa for 2 hr. Mice received IP injections of drug or saline on 3 consecutive days, starting 1 day before exposure. ABR tests were performed before exposure and 4 h, 1 day (half of the mice), and 3 d post-exposure (half of the mice). OHCs were counted for apical, middle and basal turns at 1d and 3d. Additionally, ROS-scavenging abilities of diekol and PFF-A were determined using 1,1-Diphenyl-2-picrylhydrazyl (DPPH) assay. Results showed dose-dependent scavenging ability for both compounds, and some differences in TTS between drug groups and controls that suggested some protective effects. The authors conclude that dieckol and PFF-A can prevent TTS in humans.
There is clearly a good rationale for looking at dieckol and PFF-A as potential protectants against NIHL. However, this study is disappointingly limited, making the results difficult to interpret. Overall, the introduction does not suggest a full understanding of the history and state of thinking about causes of NIHL from different levels and durations of noise exposure. There is no rationale provided for using a noise exposure of 115 dB for 2 hr, and no comparisons are made with previous studies using this exposure paradigm to provide confidence regarding measured TTS and OHC loss. It is surprising to see so much OHC loss at 1 day, and it is surprising that there was so much OHC loss in the apex on both Day 1 and Day 3. Providing measurements of PTS and OHC loss at a later time point would be most informative as to the protective effects of dieckol and PFF-A, and the authors really don’t provide a good rationale for not including PTS measures. The authors do not provide rationale for using only clicks and two frequencies, 4 kHz and 16 kHz, as stimuli for ABR measurements, nor do they explain why doses of 10 and 100 mg/kg and a 3-day injection regimine were selected for the study. Finally, the discussion does not put the results into context of previous studies, it skips some results that should be discussed (e.g., why would the low dose of dieckol have a significant effect at one time/frequency only? Why would the effects of dieckol differ from those of PFF-A? Were the frequency effects consistent with previous studies?), and it ends with a conclusion that goes far beyond the findings of this study, regarding anticipated use of dieckol and PFF-A to prevent NIHL in humans.
In addition to expanding on rationale and discussion, the authors should address the following specific points.
--The authors cite a previous study which examined the protective effects of an Ecklonia cava extract on TTS in mice and examined radical-scavenging activity using the 1,1-diphenyl-2-picrylhydrazyl assay. According to the abstract of that study, the extract exhibited “significant radical scavenging activity.” Please indicate how the assay results in the current study differs from/compares to the previous study and verify that the same data are not being used in two studies.
--Abstract:
a) please indicate level of noise exposure;
b) clarify the sentence saying that “Dieckol alleviated the ABR threshold shift”, to make it clear that the lower dose group had less TTS than either the saline+noise or high-Dieckol+noise group;
c) modify the conclusion, as it goes far beyond the findings of this study in saying dieckol and PFF-A can prevent NIHL [in humans].
--L35 NIHL is rarely “high frequency”
--L44 This sentence is unsupported and not true because it is oversimplified. The paragraph does not properly summarize the history and state of thinking about causes of NIHL from different levels and durations of noise exposure.
--L66. The current study only examines TTS in mice 1 and 3 days after noise exposure. This does not establish a model of NIHL..
--L70, The text provides the same information as Figure 2, making Figure 2 redundant and unnecessary..
--L79, why are 4 and 16 kHz in parentheses here? Is it click, 4 kHz and 16 kHz?
--L99. Clarify what the comparison group was for the “significant OHC survival in the apical turn”.
--L102, refers to “better” survival in the high-dieckol group, but were these comparisons not significant?
--L115, overstates what the results of this study show
--L201 paragraph: names of groups described in the text should match names of groups shown in Figure 5. Saline= Control? Saline = Saline + Noise? Also, numbers in Fig. 5 should be clarified, to indicate that there were n = 8 in PFFA + Noise and n = 8 in Diekol + Noise groups.
--L229, shouldn’t impedance be measured in Ohms not kW?
--L232, ABRs were also measured at 4 h post exposure—was this in all mice?
--Throughout results:
-- provide full report for all statistical results: test statistic, df, value, p, not just a p value.
--please replace the valuative terms “better” and “worse” (e.g., L84) with “lower” and “higher”, respectively.
--Fig. 3 legend—what is (1000)?
Please provide rationale for testing at only two frequencies, and why these particular frequencies (4 and 16 kHz) were selected. Also, please provide rationale for the noise exposure parameters used, and for the two doses of drugs selected.
Round 2
Reviewer 2 Report
The revised manuscript is much better. The authors have adequately addressed reviewer concerns. A few typographical errors were noticed.
Line 20: 4-kHz noise exposure (not sure what the authors mean here).
Line 75: increased
Line 125: 3 days
Line 128: So, hearing
Line 128: 1in (remove 1)
Reviewer 4 Report
The authors have adequately revised the manuscript. Minor editing of English grammar is needed.